# In Silico Evaluation of CRISPR-Based Assays for Effective Detection of SARS-CoV-2

**DOI:** 10.3390/pathogens11090968

**Published:** 2022-08-25

**Authors:** Pornchai Kaewsapsak, Naphat Chantaravisoot, Pattaraporn Nimsamer, Oraphan Mayuramart, Suwanan Mankhong, Sunchai Payungporn

**Affiliations:** 1Department of Biochemistry, Faculty of Medicine, Chulalongkorn University, Bangkok 10330, Thailand; 2Research Unit of Systems Microbiology, Faculty of Medicine, Chulalongkorn University, Bangkok 10330, Thailand; 3Center of Excellence in Systems Biology, Faculty of Medicine, Chulalongkorn University, Bangkok 10330, Thailand

**Keywords:** CRISPR, SARS-CoV-2, COVID-19, variants of concern

## Abstract

Coronavirus disease (COVID-19) caused by the SARS-CoV-2 has been an outbreak since late 2019 up to now. This pandemic causes rapid development in molecular detection technologies to diagnose viral infection for epidemic prevention. In addition to antigen test kit (ATK) and polymerase chain reaction (PCR), CRISPR-based assays for detection of SARS-CoV-2 have gained attention because it has a simple setup but still maintain high specificity and sensitivity. However, the SARS-CoV-2 has been continuing mutating over the past few years. Thus, molecular tools that rely on matching at the nucleotide level need to be reevaluated to preserve their specificity and sensitivity. Here, we analyzed how mutations in different variants of concern (VOC), including Alpha, Beta, Gamma, Delta, and Omicron strains, could introduce mismatches to the previously reported primers and crRNAs used in the CRISPR-Cas system. Over 40% of the primer sets and 15% of the crRNAs contain mismatches. Hence, primers and crRNAs in nucleic acid-based assays must be chosen carefully to pair up with SARS-CoV-2 variants. In conclusion, the data obtained from this study could be useful in selecting the conserved primers and crRNAs for effective detections against the VOC of SARS-CoV-2.

## 1. Introduction

Severe acute respiratory syndrome coronavirus 2 (SARS-CoV-2) has been spread worldwide and led to an outbreak. Until 18 July 2022, more than 567 million people have been infected, and 6 million people died due to viral infection or its complications [1]. SARS-CoV-2 is an RNA virus that relies on its unique RNA polymerase enzyme to replicate its RNA in the human host. This process can be erroneous, leading to high mutation rates in RNA viruses. In September 2020, the Alpha variant (B1.1.7) occurred, followed by the spread of the Beta variant. In late 2020, the Gamma variant (P.1) had spread, and the Delta (B.1.617.2) variant had taken over afterward. Now, Omicron variants (both BA.4 and BA.5) are dominating worldwide during 2022 [2]. The mutations in spike proteins in SARS-CoV-2 for different variants helped the virus evade host immunity. For example, the D614G mutation observed in all strains enhances viral replication, infectivity, and virion stability [3]. Other mutations occur at different frequencies and have been summarized in Figure 1.

To prevent the spread of SARS-CoV-2, molecular diagnostic techniques have been used to detect and subsequently quarantine the infected cases. The standard detection technique is reverse transcription-quantitative polymerase chain reaction (RT-qPCR). Briefly, RT-qPCR works by converting SARS-CoV-2 RNA to cDNA by reverse transcriptase and then amplifying cDNA by DNA polymerase. This technique has high sensitivity and specificity, but it takes time (more than 2 h) and requires an expensive thermocycler. Hence, there are developments in other techniques, including loop-mediated isothermal amplification (LAMP) and recombinase polymerase amplification (RPA). Both LAMP and RPA are isothermal amplification; hence, they are faster and do not require a thermocycler. However, they are prone to false positives because of non-specific amplification. Many studies have incorporated the CRISPR-Cas system to detect the amplification of the target genes. This incorporation improves the accuracy and specificity comparable to RT-qPCR [4,5]. 

Several CRISPR-based assays for SARS-CoV-2 detection have been developed and published during the past few years. Nonetheless, SARS-CoV-2 continuously evolves into several variants of concern (VOC), including Alpha, Beta, Gamma, Delta, and Omicron. Thus, the diagnostic performance of those CRISPR-based assays could be affected by the high mutation rate of SARS-CoV-2. In this systematic review, we collected the primers and crRNAs from previous works that describe the CRISPR-Cas system for SARS-CoV-2 detection up to December 2021 and then analyzed how mutations in different variants of SARS-CoV-2 could affect each assay.

## 2. Mutations in SARS-CoV-2 Variants

SARS-CoV-2 is a positive-sense single-stranded RNA virus. Its genome is about 30,000 nucleotides long, comprising 15 open reading frames (ORFs). The major ORFs encode non-structural proteins (proteases and RNA polymerase) and structural proteins, including spike (S), envelope (E), matrix (M), and nucleocapsid (N) proteins. These proteins are important for viral entry, fusion, and replication in the host cells. The S protein interacts with the angiotensin-converting enzyme 2 (ACE2) receptor and transmembrane protease serine 2 (TMPRSS2) on the host cell membrane, leading to viral entry [6,7]. Since S protein is on the viral surface membrane, it is one of the target proteins for immune response. From the SARS-CoV-2 genome database, it has been observed that this S gene has the most mutation rates, which allows it to escape the host immune system (Figure 1). D614G was the first mutation identified to increase viral replication, infectivity, and virion stability [3]. Interestingly, this mutation is found in all strains. For Alpha strain, it was first reported in the United Kingdom and contained three major mutations, del69-70, N501Y, and P681H, which enhance ACE2 receptor affinity, increase infectivity, and promote viral entry into respiratory epithelial cells, respectively [8,9,10]. The beta strain was identified in South Africa and carried E484K, K417N, and N501Y. The E484 mutation was shown to reduce the neutralization of polyclonal human plasma antibodies [11]. Similarly, the Gamma strain contained E484K, K417T, and N501Y, and it was found in Brazil. Delta strain was discovered in India and comprised G142D, L452R, E484Q, and P681R. L452R mutation was shown to help in escaping the defense mechanism [12]. Lastly, the Omicron strain was identified in South Africa. Mainly, it contains many mutations, including del69-70, T95I, G142D, del143-145, K417N, T478K, N501Y, N679K, and P681H [13]. From this variation in spike protein, the mutations on the S gene are used to classify its variants. However, there are high mutation rates in SARS-CoV-2, especially on the S gene; the detection assays for SARS-CoV-2 RNA need to be designed carefully to prevent false negative detection.

## 3. CRISPR-Cas Detection System

CRISPR-Cas system is a bacterial defense mechanism against phage. Cas nuclease can recognize and hydrolyze DNA or RNA targets using CRISPR RNA (crRNA), which has a complementary sequence to the target. This system has been applied for many applications, including gene knock-in and knock-out [14]. However, some of the Cas proteins such as Cas12a, Cas12b, and Cas13 have collateral nuclease activity, meaning that once these Cas proteins recognize the target, they can cleave nucleotide non-specifically. For Cas12a and Cas12b, the Cas protein together with its crRNA recognize double-stranded DNA (dsDNA) target and have the trans-cleavage activity for single-stranded DNA (ssDNA) [15,16]. For Cas13, the Cas protein, along with its crRNA, can hydrolyze the RNA target and later cleave single-stranded RNA (ssRNA) non-specifically [4]. This collateral activity has been utilized for nucleic acid detection (Figure 2).

For CRISPR detection, the major difference between Cas12 and Cas13 is that Cas12 only detects DNA targets with protospacer adjacent motif (PAM) sequence, while Cas13 can detect RNA targets without PAM restriction. To detect SARS-CoV-2, the CRISPR Cas12-based system needs to convert viral RNA to cDNA, followed by DNA amplification and detection by Cas12 protein. In contrast, the CRISPR Cas13-based system needs to turn RNA into cDNA for amplification with the T7 promoter. The resulting DNA was then in vitro transcribed to generate RNA for Cas13 detection. This additional in vitro transcription (IVT) could introduce more complications such as misincorporation from T7 RNA polymerase (about 0.005% error rate) [17].

In general, the limit of detection for the CRISPR-Cas system alone is in picomolar to femtomolar ranges, which is not sensitive enough to detect a few copies of targets in the attomolar range [4,18,19]. Hence, CRISPR-Cas has been combined with nucleic amplification techniques, including PCR, RPA, and LAMP, lowering the detection limit to ~10 copies per µL.

## 4. Mismatches in the Amplification Step and CRISPR-Cas Detection Step

Three major DNA amplification techniques have been used in combination with CRISPR-based assays. The first one is PCR. This approach is conventional, where the amplification consists of three main steps: denaturation at 95 °C, annealing of primers at 50–60 °C, and extension at 68–72 °C. The second one is LAMP. LAMP uses 4–8 primers binding to distinct regions. With the special design of the assay, DNA polymerase would amplify DNA products with a loop-like structure, enhancing its further amplification of numerous repeat sequences of the target. This reaction occurs at a single temperature of around 60–65 °C. The third one is recombinase-based assay, which is commercialized under recombinase polymerase amplification (RPA), recombinase-aided amplification (RAA), or enzymatic recombinase amplification (ERA). In this technique, the recombinase pairs the primers to the complementary sequence in the DNA target. The resulting displaced strand is secluded by single-stranded DNA-binding protein (SSB), followed by primer extension and DNA amplification by a strand displacing DNA polymerase. This process can happen isothermally at 37–42 °C.

Since SARS-CoV-2 has different mutations across the variants, the primers must be designed carefully to avoid mismatches. PCR has higher primer specificity than LAMP or RPA due to higher temperature during amplification [20]. However, the position of mismatches on the primer could also affect DNA amplification differently. The mismatches within the 3′ ends of primers are typically more detrimental than the internal mismatches [21,22]. 

In the CRISPR-Cas detection step, mutations of the virus can affect the binding of crRNA, resulting in compromising collateral activity. However, there is no generalizable rule to predict the effect of the mismatch positions on the trans-cleavage activity. Ooi et al. had shown that Cas12a from *Acidaminococcus* spp. with E174R/S542R/K548R (enAsCas12a) can tolerate mutation better than wild-type AsCas12a and *Lachnospiraceae* bacterium Cas12a (LbCas12a) [23]. In addition, the study also presented that the mismatches at positions 7–9 and 19 of crRNA have the most detrimental effect, while positions 1 and 17 have less effect. For Cas13, there is no systematic study on how the mismatch affects its collateral activity. However, Abudayyeh et al. reported that a single mismatch in gRNA only minimally affects *Leptotrichia shahii* Cas13a’s knockdown ability, while double mismatches in any positions in gRNA reduce knockdown efficiency dramatically [24]. For Cas13d, Wessels et al. demonstrated that the knockdown is most affected by mismatches in the seed region of gRNA between nucleotides 15–21 [25].

## 5. Mismatches in Published CRISPR-Cas Detection System

Firstly, the search was conducted electronically on PubMed from January 2020 to December 2021. The search strategy included keywords within titles or abstracts “((SARS-CoV-2) OR (severe acute respiratory syndrome coronavirus 2) OR (COVID-19) AND (CRISPR))”. Search results were imported into Covidence (https://www.covidence.org/ (accessed on 20 December 2021) for systematic review management. The abstracts of 258 candidate publications were evaluated by three reviewers focusing on the articles related to the molecular techniques (amplification-free or PCR or LAMP or RPA) combined with a CRISPR-based assay for SARS-CoV-2 detection. Review articles and irrelevant studies were excluded during the evaluation process. Finally, the primer and crRNA sequences were retrieved from the full texts of 53 candidate articles (Appendix A). Genome sequences of SARS-CoV-2, including Alpha, Beta, Gamma, Delta, and Omicron variants, were multiple aligned and then analyzed for primer and crRNA binding sites. The mismatches or deletions within the primers or crRNA binding sites were illustrated in Appendix A.

For our compilation, 33% (4/12) of PCR primer pairs have mismatches for different variants, while there are 21% (7/33) and 54% (26/48) of LAMP and RPA/RAA primers containing mismatches, respectively (Table 1). However, in the PCR-based assay, only one of the crRNAs (1/12) has mismatches with the Omicron variant. For LAMP-based assay, 6% (2/34) of the crRNAs have mismatches with Alpha, Beta, and Omicron variants. For RPA/RAA-based assay, 23% (11/48) of the crRNAs have mismatches with Alpha, Gamma, Delta, and Omicron variants.

Considering which viral genes have the most mismatches in amplification primers, we found that the S gene has the most mismatches (82%), while the E and N genes have 50% and 39% mismatches, respectively. This result agrees that the S gene has the highest variation among the genes in the SARS-CoV-2 genome. Strain-wise, the primer pairs have the lowest mismatches for gamma strain (11%), while mismatches in other strains (alpha, beta, delta, and Omicron) are almost double (18%, 21%, 19%, and 22%, respectively). 

Next, we examined CRISPR-Cas reactions, and we found that 15% of the CRISPR-Cas reactions have mismatches between the viral genome and crRNA. Dividing by type of Cas (Cas12 versus Cas13), crRNAs in Cas12 has slightly lower mismatch percentages than crRNA in Cas13 (14% versus 20%, respectively). Strain-wise, crRNAs have the most mismatches with Delta and Omicron (7%), while Beta, Gamma, and Alpha have 2%, 2%, and 4% mismatches, respectively. Considering the target genes, the S gene has the most mismatches with crRNAs (25%), while E, ORF1ab, and N genes have mismatches at 0%, 15%, and 17%, respectively. This suggests that crRNAs of the E gene are robust for SARS-CoV-2 detection. Hence, the E gene could have potential use as a universal detection gene, while the S gene could be beneficial for designing assays to differentiate viral strains. 

## 6. Conclusions

During the era of the SARS-CoV-2 pandemic, early diagnosis and screening are crucial to prevent the spread of the virus. However, due to the rapid mutation rate of the viral genome, molecular detection assays, such as RT-qPCR and CRISPR-Cas systems, that rely on nucleotide sequence need to be implemented cautiously. For the CRISPR-Cas system, we found that one-third of the primer sets in the reported studies and 15% of those assays have mismatches with crRNAs. Thus, for the current and upcoming variants of SARS-CoV-2, the primer and crRNA sequences need to be aligned with the current spreading viral genome to ensure the accuracy and specificity of the result.

## Figures and Tables

**Figure 1 pathogens-11-00968-f001:**
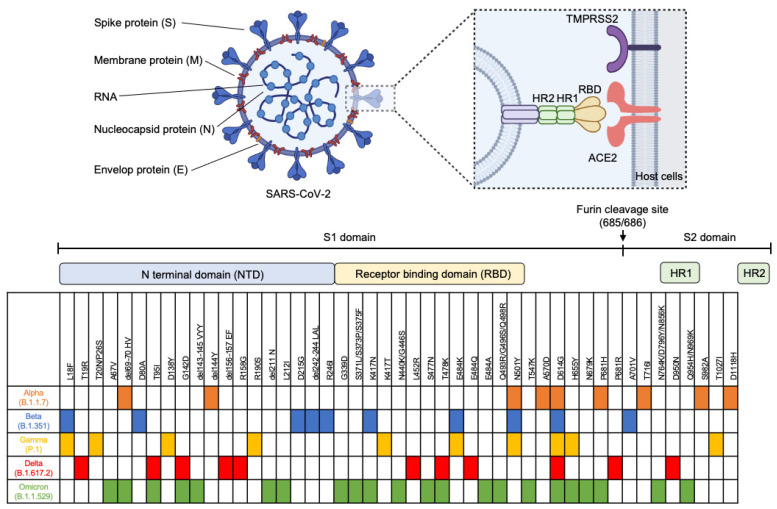
Structure of SARS-CoV-2 and summary of key mutations on spike proteins in SARS-CoV-2 variants of concern. ACE2: Angiotensin-converting enzyme 2 receptor. HR1: Heptad repeat 1. HR2: Heptad repeat 2. RBD: Receptor binding domain. TMPRSS2: Transmembrane protease serine 2. This figure was created with BioRender.com.

**Figure 2 pathogens-11-00968-f002:**
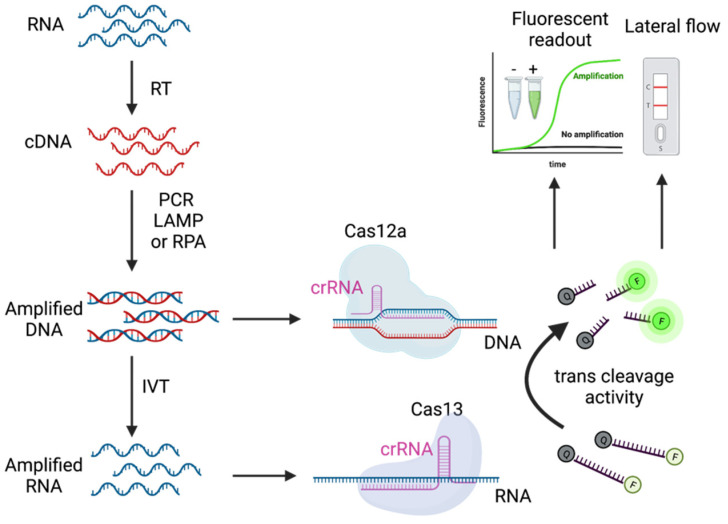
CRISPR-Cas detection system for SARS-CoV-2 RNA. Extracted RNAs from SARS-CoV-2 RNA were reverse-transcribed to cDNA and subsequently amplified by either PCR, LAMP, or RPA. The amplified DNA together with matched crRNA can activate collateral activity in the Cas12a enzyme. For Cas13, the amplified DNA is used as the IVT template to generate multiple copies of RNA targets, which can activate the collateral activity of Cas13 with matched crRNA. The collateral activity of Cas enzyme can cleave nucleotide reporter for fluorescent readout or lateral flow. This figure was created with BioRender.com.

**Table 1 pathogens-11-00968-t001:** Mismatches in primers and crRNAs in different detection assays.

Title	Target	Amp	Primers Match for Variants of Concern					Cas	crRNA	Remark	Ref
			α	β	γ	δ	O				
A CRISPR-based and post-amplification coupled SARS-CoV-2 detection with a portable evanescent wave biosensor	S	-	-	-	-	-	-	Cas13a	✓		[26]
	N	-	-	-	-	-	-	Cas13a	✓		
	Orf1ab	-	-	-	-	-	-	Cas13a	MM1/MM2	MM1:	
										β, γ, δ, O	
										MM2: α	
A Novel Miniature CRISPR-Cas13 System for SARS-CoV-2 Diagnostics	N	LAMP	✓	✓	✓	✓	✓	mCas13	✓		[27]
A one-step, one-pot CRISPR nucleic acid detection platform (CRISPR-top): Application for the diagnosis of COVID-19	Orf1ab	LAMP	✓	✓	✓	✓	✓	Cas12b	✓		[28]
A Saliva-Based RNA Extraction-Free Workflow Integrated With Cas13a for SARS-CoV-2 Detection	N	LAMP	✓	✓	✓	MM1	✓	Cas12b	✓		[29]
	S	RPA	✓	✓	✓	✓	✓	Cas13a	✓		
	Orf1ab	RPA	✓	✓	✓	✓	✓	Cas13a	✓		
A Scalable, Easy-to-Deploy Protocol for Cas13-Based Detection of SARS-CoV-2 Genetic Material	N1	PCR	✓	MM1	✓	✓	✓	Cas13a	MM1	Omicron	[30]
	N2	PCR	✓	✓	✓	✓	✓	Cas13a	✓		
	N3	PCR	✓	✓	✓	✓	✓	Cas13a	✓		
A smartphone-based visual biosensor for CRISPR-Cas powered SARS-CoV-2 diagnostics	N	PCR	MM4	MM1	MM3	MM1	MM3	Cas12a	✓		[31]
A smartphone-read ultrasensitive and quantitative saliva test for COVID-19	Orf1ab	PCR	✓	✓	✓	✓	✓	Cas12a	✓		[32]
	N	PCR	MM4	MM1	MM3	MM1	MM3	Cas12a	✓		
A Thermostable Cas12b from Brevibacillus Leverages One-pot Detection of SARS-CoV-2 Variants of Concern	N	LAMP	✓	✓	✓	✓	✓	Cas12b	✓		[33]
Amplification-free detection of SARS-CoV-2 with CRISPR-Cas13a and mobile phone microscopy	N	-	-	-	-	-	-	Cas13a	✓	All three crRNAs are perfect match	[34]
An engineered CRISPR-Cas12a variant and DNA-RNA hybrid guides enable robust and rapid COVID-19 testing	S	LAMP	✓	MM1	✓	✓	Del3, Ins9	Cas12a	✓		[23]
Application of the amplification-free SERS-based CRISPR/Cas12a platform in the identification of SARS-CoV-2 from clinical samples	N	-	-	-	-	-	-	Cas12a	✓		[35]
Clinical validation of a Cas13-based assay for the detection of SARS-CoV-2 RNA	S	RPA	✓	DEL9	✓	✓	✓	Cas13a	✓		[36]
	Orf1ab	RPA	✓	✓	✓	✓	✓	Cas13a	DEL3	Omicron	
	Orf1b	RPA	✓	✓	✓	✓	✓	Cas13a	✓		
	N	RPA	MM1	✓	✓	✓	✓	Cas13a	MM1	Delta	
Contamination-free visual detection of SARS-CoV-2 with CRISPR/Cas12a: A promising method in the point-of-care detection	E	LAMP	✓	✓	✓	✓	MM1	Cas12a	✓		[37]
	Orf1ab	LAMP	✓	✓	✓	✓	✓	Cas12a	✓		
	N	LAMP	MM1	✓	✓	✓	✓	Cas12a	MM1	Alpha	
CRISPR/Cas12a Technology Combined with RT-ERA for Rapid and Portable SARS-CoV-2 Detection	N	ERA	✓	✓	✓	✓	✓	Cas12a	✓		[38]
	Orf1ab	ERA	✓	✓	✓	✓	✓	Cas12a	✓		
CRISPR/Cas12a-mediated gold nanoparticle aggregation for colorimetric detection of SARS-CoV-2	N	LAMP	✓	✓	✓	✓	✓	Cas12a	✓		[39]
	E	LAMP	✓	✓	✓	✓	✓	Cas12a	✓		
CRISPR-Cas12-based detection of SARS-CoV-2	N	LAMP	✓	✓	✓	✓	✓	Cas12a	✓		[40]
	E	LAMP	✓	✓	✓	✓	✓	Cas12a	✓		
Detection of Infectious Viruses Using CRISPR-Cas12-Based Assay	S	RPA	✓	MM1	MM1	MM1	MM3	Cas12a	✓		[41]
Detection of SARS-CoV-2 by CRISPR/Cas12a-Enhanced Colorimetry	E	RPA	✓	MM1	✓	✓	MM1	Cas12a	✓		[42]
	Orf1ab	RPA	✓	✓	✓	✓	✓	Cas12a	✓		
	N1	RPA	MM1	✓	✓	✓	✓	Cas12a	✓		
	N2	RPA	✓	✓	✓	✓	✓	Cas12a	✓		
Detection of severe acute respiratory syndrome coronavirus 2 and influenza viruses based on CRISPR-Cas12a	S1	RPA	✓	✓	✓	MM1, DEL6	MM1	Cas12a	✓		[43]
	S2	RPA	✓	✓	✓	MM1	MM2	Cas12a	MM3	Omicron	
Detection of the SARS-CoV-2 D614G mutation using engineered Cas12a guide RNA	E	RPA	✓	MM1	✓	✓	MM1	Cas12a	✓		[44]
Development and evaluation of a rapid CRISPR-based diagnostic for COVID-19	Orf1ab	RPA	✓	✓	✓	✓	✓	Cas13a	✓		[45]
Development of a Broadly Applicable Cas12a-Linked Beam Unlocking Reaction for Sensitive and Specific Detection of Respiratory Pathogens Including SARS-CoV-2	S	RPA	DEL6	✓	✓	✓	MM1, DEL6	Cas12a	✓		[46]
Development of a Rapid and Sensitive CasRx-Based Diagnostic Assay for SARS-CoV-2	S3	RPA	MM1	✓	✓	✓	✓	Cas13d	✓		[47]
	N1	RPA	✓	✓	✓	✓	✓	Cas13d	✓		
Digital CRISPR/Cas-Assisted Assay for Rapid and Sensitive Detection of SARS-CoV-2	N	RPA	MM4	MM1	MM5	MM1	MM3	Cas12a	MM1	Delta: One crRNA has MM. The other has PM.	[48]
Electric field-driven microfluidics for rapid CRISPR-based diagnostics and its application to detection of SARS-CoV-2	N	LAMP	✓	✓	✓	✓	✓	Cas12a	✓		[49]
	E	LAMP	✓	✓	✓	✓	✓	Cas12a	✓		
Enhancement of trans-cleavage activity of Cas12a with engineered crRNA enables amplified nucleic acid detection	N1	LAMP	MM1	MM2	✓	DEL6	✓	Cas12a	✓		[50]
	N2	LAMP	✓	✓	✓	✓	✓	Cas12a	✓		
Fluorescence polarization system for rapid COVID-19 diagnosis	N1	RPA	MM4	MM1	MM5	MM1	MM3	Cas12a	MM1	Delta	[51]
	N2	RPA	✓	✓	✓	✓	✓	Cas12a	✓		
Instrument-free, CRISPR-based diagnostics of SARS-CoV-2 using self-contained microfluidic system	N	RPA	✓	✓	✓	✓	✓	Cas12a	✓		[52]
iSCAN: An RT-LAMP-coupled CRISPR-Cas12 module for rapid, sensitive detection of SARS-CoV-2	E	LAMP	✓	MM1	✓	✓	✓	Cas12a	✓		[53]
								Cas12b	✓		
Isothermal Amplification and Ambient Visualization in a Single Tube for the Detection of SARS-CoV-2 Using Loop-Mediated Amplification and CRISPR Technology	N	LAMP	✓	✓	✓	✓	✓	Cas12a	✓		[54]
	E	LAMP	✓	✓	✓	✓	✓	Cas12a	✓		
MeCas12a, a Highly Sensitive and Specific System for COVID-19 Detection	E	RPA	✓	MM1	✓	✓	MM1	Cas12a	✓		[55]
Minimally instrumented SHERLOCK (miSHERLOCK) for CRISPR-based point-of-care diagnosis of SARS-CoV-2 and emerging variants	N	RPA	MM1	✓	✓	✓	✓	Cas12a	✓		[56]
One-tube SARS-CoV-2 detection platform based on RT-RPA and CRISPR/Cas12a	RdRp	RPA	✓	✓	✓	✓	✓	Cas12a	MM1	Delta	[57]
	N	RPA	MM4	MM1	MM5	MM1	MM3	Cas12a	✓		
Point-of-care CRISPR-Cas-assisted SARS-CoV-2 detection in an automated and portable droplet magnetofluidic device	N	RPA	MM4	MM1	MM5	MM1	MM3	Cas12a	MM1	Delta	[58]
Point-of-care testing for COVID-19 using SHERLOCK diagnostics	N	LAMP	✓	✓	✓	✓	✓	Cas12b	✓		[59]
Rapid and sensitive detection of COVID-19 using CRISPR/Cas12a-based detection with naked eye readout, CRISPR/Cas12a-NER	Orf1a	RPA	✓	✓	✓	✓	MM1	Cas12a	✓		[60]
	Orf1b	RPA	✓	✓	✓	MM1	✓	Cas12a	✓		
	E	RPA	✓	MM1	✓	✓	MM1	Cas12a	✓		
	N	RPA	✓	✓	✓	✓	✓	Cas12a	MM1	Alpha	
Rapid and Sensitive Detection of SARS-CoV-2 Using Clustered Regularly Interspaced Short Palindromic Repeats	M	RPA	✓	✓	✓	✓	✓	Cas12a	✓		[61]
	N2	RPA	✓	✓	✓	✓	✓	Cas12a	✓		
	S2	RPA	✓	✓	✓	✓	✓	Cas12a	MM1	Gamma, Omicron	
Rapid Detection of 2019 Novel Coronavirus SARS-CoV-2 Using a CRISPR-based DETECTR Lateral Flow Assay	N2	RPA	✓	✓	✓	✓	✓	Cas12a	✓		[62]
	E	RPA	✓	✓	✓	✓	✓	Cas12a	✓		
	N	LAMP	✓	✓	✓	✓	✓	Cas12a	✓		
	E	LAMP	✓	✓	✓	✓	✓	Cas12a	✓		
Rapid detection of SARS-CoV-2 with CRISPR-Cas12a	Orf1ab	LAMP	✓	✓	✓	MM1	✓	Cas12a	✓		[63]
	N	LAMP	MM3	MM1	MM5	MM1	MM3	Cas12a	✓		
Rapid SARS-CoV-2 testing in primary material based on a novel multiplex RT-LAMP assay	Orf1a	LAMP	✓	✓	✓	✓	✓	Cas13a	✓		[64]
	N	LAMP	✓	✓	✓	✓	✓	Cas13a	✓		
	ORF7a	LAMP	✓	✓	✓	✓	✓	Cas13a	✓		
	ORF3a	LAMP	✓	✓	✓	✓	✓	Cas13a	✓		
	Orf1ab	RPA	✓	✓	✓	✓	✓	Cas13a	DEL3	Omicron	
	S	RPA	✓	DEL9	✓	✓	✓	Cas13a	✓		
Reverse Transcription Recombinase Polymerase Amplification Coupled with CRISPR-Cas12a for Facile and Highly Sensitive Colorimetric SARS-CoV-2 Detection	Orf1ab	RPA	✓	✓	✓	✓	MM1	Cas12a	✓		[65]
	N	RPA	MM4	MM1	MM3	MM1	MM3	Cas12a	✓		
SARS-CoV-2 detection with CRISPR diagnostics	RdRp	RAA	MM1	✓	✓	MM1	✓	Cas12b	✓		[66]
SARS-CoV-2 Direct Detection Without RNA Isolation With Loop-Mediated Isothermal Amplification (LAMP) and CRISPR-Cas12	N	LAMP	✓	✓	✓	✓	✓	Cas12a	✓		[67]
SARS-CoV-2 RNA Detection by a Cellphone-Based Amplification-Free System with CRISPR/CAS-Dependent Enzymatic (CASCADE) Assay	Orf1ab	PCR	✓	✓	✓	✓	✓	Cas12a	✓		[68]
Sensitive and Easy-Read CRISPR Strip for COVID-19 Rapid Point-of-Care Testing	N	RAA	✓	✓	✓	✓	✓	Cas13a	✓		[69]
Sensitive and rapid on-site detection of SARS-CoV-2 using a gold nanoparticle-based high-throughput platform coupled with CRISPR/Cas12-assisted RT-LAMP	N	LAMP	✓	✓	✓	✓	✓	Cas12a	✓		[70]
Sensitive tracking of circulating viral RNA through all stages of SARS-CoV-2 infection	Orf1ab	PCR	✓	✓	✓	✓	✓	Cas12a	✓		[71]
Streamlined inactivation, amplification, and Cas13-based detection of SARS-CoV-2	Orf1ab	RPA	✓	✓	✓	✓	✓	Cas13a	✓		[72]
Ultra-sensitive and high-throughput CRISPR-powered COVID-19 diagnosis	Orf1ab	PCR	✓	✓	✓	✓	✓	Cas12a	✓		[73]
	N	PCR	MM4	MM1	MM3	MM1	MM3	Cas12a	✓		
Ultrasensitive and visual detection of SARS-CoV-2 using all-in-one dual CRISPR-Cas12a assay	N	RPA	MM4	MM1	MM5	MM1	MM3	Cas12a	MM1	Delta: use 3 crRNAs and one crRNA has one MM	[74]
UnCovid: A versatile, low-cost, and open-source protocol for SARS-CoV-2 RNA detection	Orf1ab	PCR	✓	✓	✓	✓	✓	Cas12a	✓		[75]
	N	PCR	✓	✓	✓	✓	✓	Cas12a	✓		
Universally Stable and Precise CRISPR-LAMP Detection Platform for Precise Multiple Respiratory Tract Virus Diagnosis Including Mutant SARS-CoV-2 Spike N501Y	N	LAMP	✓	✓	✓	✓	✓	Cas12a	✓		[76]
	RdRp	LAMP	✓	✓	✓	✓	✓	Cas12a	✓		
	S	LAMP	✓	✓	✓	MM1	✓	Cas12a	MM1, MM3	Alpha and Beta strains have MM1. Omicron strain has MM3.	
Unlocking SARS-CoV-2 detection in low- and middle-income countries	N	LAMP	✓	✓	✓	✓	✓	Cas12a	✓		[77]
	Orf1ab	LAMP	✓	✓	✓	✓	✓	Cas12a	✓		

α, β, γ, δ, and O mean Alpha, Beta, Gamma, Delta, and Omicron variants, respectively. MMn = n number of mismatches. INSn = n number of inserted nucleotide. DELn = n number of deleted nucleotides.

## Data Availability

Not applicable.

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
