# Peer review of "In Silico Evaluation of CRISPR-Based Assays for Effective Detection of SARS-CoV-2"

_pathogens, 2022, doi:10.3390/pathogens11090968_

Round 1
Reviewer 1 Report
Authors have write this manuscript very nicely. Contains are good and fit for this journal. Authors should have to add or rewrite section 2 (Mutations in SARS-CoV-2 variants). It is very superficial information. Kindly add more deep information about it and try to add mutation sites. Figure format will looks good.
Author Response
Reviewer 1:
Authors have write this manuscript very nicely. Contains are good and fit for this journal. Authors should have to add or rewrite section 2 (Mutations in SARS-CoV-2 variants). It is very superficial information. Kindly add more deep information about it and try to add mutation sites. Figure format will looks good.
Response: We would like first to thank you for your valuable suggestion. As advised, we have discussed key mutations for each variant (lines 85-96) and briefly mentioned their potential roles. In addition, we have included the figure of the SARS-CoV-2 structure to aid understanding of the location of mutations on the spike proteins.
Reviewer 2 Report
In this study, the authors analyzed the mutations in different variants of concern (VOC), including Alpha, Beta, Gamma, Delta, and Omicron strains which may affect the previously reported primers and crRNAs used in the CRISPR-Cas system. They found that over 40% of the primer sets and 15% of the crRNAs contain mismatches. Results from this study could be useful in selecting the conserved primers and crRNAs for effective detections against the VOC of SARS-CoV-2.
Several suggestions:
1. Line 31, [reverse transcriptase] should be changed to [RNA polymerase].
2. Please add references after these sentences. [line 30, ……due to viral infection or its complications]; [line 36, ……dominating worldwide during 2022]; [line 53, ……. comparable to RT-qPCR].
3. This study found mismatches between the primers (or crRNAs) and sequences of different VOCs. These mismatches may or may not affect the detection of SARS-CoV-2. [How do you know these mismatches will definitely affect the detection?]. Therefore, it is better to rephrase the following sentence [Here we analyzed how mutations in different variants of concern (VOC), including Alpha, Beta, Gamma, Delta, and Omicron strains affected the previously reported primers and crRNAs used in the CRISPR-Cas system.].
Author Response
Reviewer 2:
In this study, the authors analyzed the mutations in different variants of concern (VOC), including Alpha, Beta, Gamma, Delta, and Omicron strains which may affect the previously reported primers and crRNAs used in the CRISPR-Cas system. They found that over 40% of the primer sets and 15% of the crRNAs contain mismatches. Results from this study could be useful in selecting the conserved primers and crRNAs for effective detections against the VOC of SARS-CoV-2.
Several suggestions:
- Line 31, [reverse transcriptase] should be changed to [RNA polymerase].
Response: Thank you for pointing this out. We have changed “reverse transcriptase” to “RNA polymerase” accordingly (line 32).
- Please add references after these sentences. [line 30, ……due to viral infection or its complications]; [line 36, ……dominating worldwide during 2022]; [line 53, ……. comparable to RT-qPCR].
Response: We have added the references as suggested in lines 32, 37, and 57 of the revised manuscript.
- This study found mismatches between the primers (or crRNAs) and sequences of different VOCs. These mismatches may or may not affect the detection of SARS-CoV-2. [How do you know these mismatches will definitely affect the detection?]. Therefore, it is better to rephrase the following sentence [Here we analyzed how mutations in different variants of concern (VOC), including Alpha, Beta, Gamma, Delta, and Omicron strains affected the previously reported primers and crRNAs used in the CRISPR-Cas system.].
Response: We are sorry for the confusion. We have revised the sentence to “Here we analyzed how mutations in different variants of concern (VOC), including Alpha, Beta, Gamma, Delta, and Omicron strains, could introduce mismatches to the previously reported primers and crRNAs used in the CRISPR-Cas system.” (line 20). We hope this would emphasize only the occurrence of the mismatches on the primers and crRNAs, not their effects on the assay’s efficacy.
Reviewer 3 Report
Kaewsapsak et al. reviewed vast majority of articles published in the past few year looking at the diagnostic for SARS-CoV-2/variants and report some of the mismatches with primers/gRNA in these assay against some of the variants. This is very important contribution to the field as they have summarized most of the study performed till date. Anecdotally, I have seen even 1 nucleotide difference in gRNA/primers affects the result.
The authors have written and presentation the summarized data beautifully! I have rarely seen reviews written in such succinct way with easily accessible information to the reader. I fully support the publication of this review article. Kudos to the authors, thank you for doing this.
Author Response
Kaewsapsak et al. reviewed vast majority of articles published in the past few year looking at the diagnostic for SARS-CoV-2/variants and report some of the mismatches with primers/gRNA in these assay against some of the variants. This is very important contribution to the field as they have summarized most of the study performed till date. Anecdotally, I have seen even 1 nucleotide difference in gRNA/primers affects the result.
The authors have written and presentation the summarized data beautifully! I have rarely seen reviews written in such succinct way with easily accessible information to the reader. I fully support the publication of this review article. Kudos to the authors, thank you for doing this.
Response: Thank you for your compliment. To make the paper more vivid and informative, we have included the structure of SARS-CoV-2 in Figure 1.